# LMEye: An Interactive Perception Network for Large Language Models

## Abstract

Training a Multimodal Large Language Model (MLLM) from scratch, like GPT-4, is resource-intensive. Regarding Large Language Models (LLMs) as the core processor for multimodal information, our paper introduces LMEye, a human-like eye with a play-and-plug interactive perception network, designed to enable dynamic interaction between LLMs and external vision information. Previous methods incorporate visual information into LLMs with a simple visual mapping network or Q-former from BLIP-2. Such networks project the image feature once yet do not consider the interaction between the image and the human input query. Hence, the obtained visual information without being connected to human intention may be inadequate for LLMs to generate intention-following responses, which we refer to as static visual information. LMEye addresses this issue by allowing the LLM to request the desired visual information aligned with various human instructions, which we term as the dynamic visual information interaction. Specifically, LMEye consists of a simple visual mapping network to provide the basic perception of an image for LLMs. It also contains additional modules responsible for acquiring requests from LLMs, performing request-based visual information interaction, and transmitting the resulting interacted visual information to LLMs, respectively. In this way, LLMs act to understand the human query, deliver the corresponding request to the request-based visual information interaction module, and generate the response based on the interleaved multimodal information. We evaluate LMEye through extensive experiments on some multimodal benchmarks, demonstrating that it significantly improves the zero-shot performance on various multimodal tasks compared to previous methods, with less parameters.

## 1 Introduction

Vision-Language Models (VLMs) (Alayrac et al., 2022; Wang et al., 2022a) trained on a massive amount of image-text data have shown impressive results in various multimodal understanding and generation tasks. However, training a MLLM (e.g., Flamingo (Alayrac et al., 2022), Kosmos-1 (Huang et al., 2023), and GPT-4 (OpenAI, 2023b)) from scratch is resource-intensive. To alleviate this issue, previous open-source efforts (Merullo et al., 2023; Li et al., 2023c; Driess et al., 2023; Guo et al., 2023) present that we can construct a MLLM based on the text-only large language model (LLM) through transforming the visual information (obtained by frozen pretrained visual encoders (Radford et al., 2021; Dosovitskiy et al., 2020)) into the representation space of LLM. By doing so, LLMs are capable of understanding visual information and performing multimodal human-machine interaction. Significantly, the whole training process is parameter efficient since it only needs to optimize a few parameters of the vision-to-language feature transformer, similar to popular prefix or prompt tuning approaches (Li & Liang, 2021; Jia et al., 2022).

Recent work (Merullo et al., 2023) demonstrates that a learnable linear mapping network can allow LLMs to incorporate the basic global perception information of an image. Different from common VLMs, e.g., Oscar (Li et al., 2020) and OFA (Wang et al., 2022a), MLLMs constructed in this way usually perform multimodal generation well (Merullo et al., 2023) because LLMs are capable of powerful contextual understanding, reasoning, and generating capabilities. To step forward this direction, Koh et al. (2023) present the model FROMAGe, where they freeze the LLM and visual encoder and fine-tune several linear mapping layers to achieve cross-modality information interactions. It realizes strong zero-shot performances on the contextual image retrieval and multimodal dialogue

tasks. Li et al. (2023c) propose BLIP-2 with a lightweight Querying Transformer to bridge the vision and language semantic gap for frozen image encoders and large language models. In addition, the multimodal instruction-following tuning approach is recently introduced by Liu et al. (2023a) and Zhu et al. (2023) to advance the multimodal interaction capability of LLMs, which show supervisor performances on various multimodal scenarios.

However, for previous methods such as BLIP-2 and FROMAGe, the visual feature fed into LLMs is transformed once via the visual mapping network and does not interact with human input queries, which we term as static visual information. Hence, the language model may not obtain the adequate visual information for various queries. To address this issue, we present a human-like eye with interactive perception network for LLMs, namely LMEye, which allows LLMs to request the desired visual information aligned with various human instructions. From the Agent's perspective, we consider LLMs as the core processor of complex information and do not modify the structure of LLMs. Otherwise, it may the risk of degenerating their original performances on NLP tasks, similar to Driess et al. (2023), thereby weakening the generalization of LLMs. Concretely, LMEye mainly consists of two stages: 1) the first one provides the basic perception information of an image for LLMs, called feature alignment. We adopt a widely-used visual mapping network Q-Former from BLIP-2 to achieve it. 2) another one is responsible for acquiring the request from LLMs, performing request-based visual information interaction, and transmitting the resulting interacted visual information to LLMs. We introduce a Request-based Visual Information Interaction module (RVII) to perform such dynamic interaction between LLMs and visual information. By doing so, LLMs first understand human queries and basic information of an image, then send a request to obtain additionally desired visual information, finally generate the instruction-following response based on basic image information, text instruction, and interacted visual information.

In summary, the contributions of our proposed LMEye lie in the following three-fold.

- Regarding LLMs as multimodal information processor, we propose an interactive perception network to make them incorporate the desired visual information for various human queries. LLMs act to understand the human query, deliver the corresponding request to the request-based visual information interaction module, and generate the response based on the interleaved multimodal information. The whole training process is parameter-efficient.

- LMEye could achieve superior multimodal understanding and reasoning performances on two multimodal evaluation benchmarks (MMBench and SEED-Bench) with less parameters (4.4B), compared to other MLLMs (> 7B).

- Ablation studies show that the proposed method significantly improves zero-shot mutlimodal performances for various scales and types of LLMs, e.g., exact gain by 5.0% on OK-VQA for LMEye (BLIP-2) vs. BLIP-2, and exact gain by 20% on VQA with long answer for LMEye (LLaMA-7b) vs. LLaVA (Vicuna-7b).

## 2 RELATED WORK

**Vision-assisted LLMs**. Different vision-language models (Alayrac et al., 2022; Wang et al., 2022b; Huang et al., 2023) which are trained from scratch with large-scale image-text pairs, vision-assisted LLMs is based on a pre-trained large language model, allowing it to understand visual information and be able to process multimodal information. They usually apply the recently proposed prefix-tuning (Jia et al., 2022; Li & Liang, 2021) or adapter-based (Zhang et al., 2023b) tuning methods to fine-tune the language model on specific multimodal tasks, so that they can be competent for some multimodal scenarios. For instance, Zhu et al. (2022) utilized the text-to-image technical to generate the image and infused the visual information into the language model for multimodal text generation. Koh et al. (2023) explored using LLMs for image-text retrieval and multimodal text-image interaction. To step forward this direction, BLIP-2 (Li et al., 2023c) employs a Flan-T5 (Chung et al., 2022) or OPT (Zhang et al., 2022) with a Q-Former to efficiently align visual features with the language model. Most recently, PaLM-E (Driess et al., 2023), featuring 562 billion parameters, has been developed to integrate real-world continuous sensor modalities into an LLM, thereby establishing a connection between real-world perceptions and human languages. To conclude, previous works demonstrate that it is a potential research direction for enabling frozen LLMs to handle multimodal information.

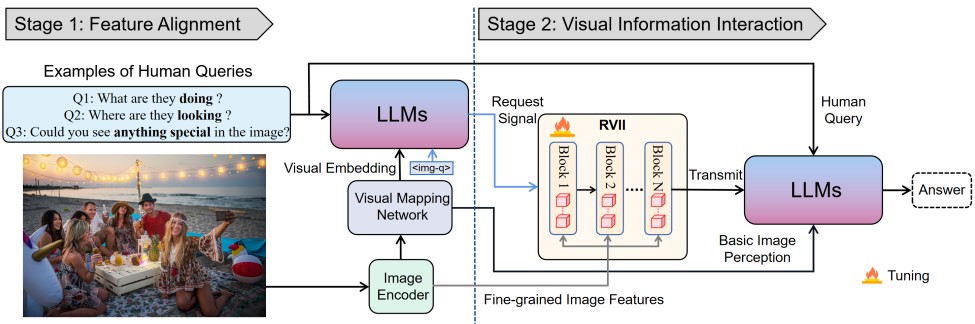

Figure 1: Illustration of the overall architecture of LMEye. Image Encoder, Visual Mapping Network, and LLMs are frozen during training. RVII represents the Request-based Visual Information Interaction module.

**Multimodal Instruction-following Tuning**. Advances in instruction-tuning text-only LLMs (Ouyang et al., 2022; Muennighoff et al., 2022; Chung et al., 2022) have achieved impressive performance on the NLP tasks and human-machine interaction scenarios, such as Flan-T5, Bloomz (Muennighoff et al., 2022), and ChatGPT. Recently, some researchers have explored using multimodal instruction data to fine-tune pre-trained LLMs to improve their multimodal human-machine interaction capability. Liu et al. (2023a) employs the GPT-4 to produce the multimodal instruction data and fine-tune the language model LLaMA (Touvron et al., 2023) on the synthetic multimodal instruction-following dataset. Zhu et al. (2023) also construct a well-aligned multimodal instruction-following dataset to fine-tune a robust instruction-tuned language model (Vicuna), and it achieves superior performance on open-domain multimodal dialogue. Zhang et al. (2023b) present a lightweight adaption method to efficiently fine-tune LLaMA into an instruction-following model. In this paper, we introduce various multimodal instruction data to make LMEye adapt to open-domain multimodal scenarios.

**Vision Tools for LLMs**. A recent line of research (Mialon et al., 2023; Qin et al., 2023) investigates ways to enhance the performance of LLMs by enabling them to access external tools such as vision foundation models, search engines, or other APIs for solving complex problems. This approach broadens the scope of LLMs in processing information of varying complexities. For example, Toolformer (Schick et al., 2023) empowers LLMs to decide which APIs to use, when to use them, what arguments to pass, and how to incorporate the resulting information into text generation. Low-code LLM (Cai et al., 2023) uses six simple low-code visual programming interactions, such as clicking, dragging, or text editing, to achieve more controlled and reliable responses. In contrast, Lu et al. (2023) propose a plug-and-play compositional reasoning framework Chameleon that augments LLMs to address complex challenges, such as using off-the-shelf vision models. Wu et al. (2023) introduce visual ChatGPT, which designs a set of prompts to incorporate visual model information into ChatGPT, considering models with multiple inputs/outputs and those that require visual feedback. Unlike the above pipeline approaches, our work focuses on end-to-end interaction framework between LLMs and visual information.

## 3 LMEYE: INTERACTIVE PERCEPTION NETWORK

### 3.1 ARCHITECTURE

As shown in Figure 1, the overall architecture of LMEye contains two stages, which are response for different functions. Given an image $I$ and one human's query $X = (x_1, ..., x_M)$, where $x_i$ represents the $i$-th token of the human query input to LLMs, we obtain the global and fine-grained image features $h_I = (h_g^I, h_1^I, ..., h_{256}^I)$ via the pretrained visual encoder of BLIP-2. Meanwhile, a learnable special token $$ is added in the word embedding table of LLMs as the input position tag of image feature.

**Feature Alignment**. We employ frozen Q-former from BLIP-2 and a learnable linear project layer as the visual mapping network to project the basic image feature into the language embedding space, denoted to $f(h_I)$. By doing so, LLMs can obtain the basic perception information of an image,

which is added with the presentation of token **. We also add another special token ** at the end of the image and human query sequence to capture the whole encoding information of image and human query. Hence, as the left part shown in Figure 1, we can obtain the first input sequence of LLMs by (**, $f(h_I)$, $X$, **) $\rightarrow$ ($h_{img}$, $h_X$, $h_{img\text{-}q}$), where $h_{img}$ refers to the addition of $f(h_I)$ and the token representation of **. $h_X$ and $h_{img\text{-}q}$ are the corresponding word encoding representations of $X$ and **. The final output of the token ** at the last layer of LLMs is expected to contain the semantic meaning of human query, which is present to $h_r \in \mathrm{R}^{1 \times d_L}$, where $d_L$ refers to the hidden state size of LLM, since previous works (OpenAI, 2023a;b; Touvron et al., 2023) have shown that recent LLMs have been able to understand various human languages. In addition, $h_r$ may also contain the image content via the self-attention mechanism of LLMs, yet we argue that text-only LLMs without pretraining on multimodal data cannot incorporate visual information (Merullo et al., 2023) well like powerful pretrained multimodal image and language models. To facilitate LLMs incorporating the desired visual information aligned with a human query, instead of optimizing the parameters of LLMs (with full-parameter or the Low-rank adaptation Hu et al. (2021)) on specific data, like LLaVA (Liu et al., 2023a) and mPLUG-Owl (Ye et al., 2023), we perform the interaction between human query and visual information outside LLMs. In this way, LLMs could still maintain their original powers and generalization on natural language tasks because the structure and parameters of LLMs are not changed.

**Request-based Visual Information Interaction (RVII)**. First, we apply a linear project layer to map the above hidden state $h_r$ into the space of the following information interactive module, denoted to $h_R \in \mathrm{R}^{Q \times d_{RV}}$, where $Q$ and $d_{RV}$ refer to the length of request vector and the hidden size of RVII, respectively. We regard that this process is acquiring the request signal from LLMs, e.g., as the example shown in Figure 1, LLMs encoding request information from human query: "*Q1: What are they doing in the image? Q3: Could you see anything special in the image?*".

After gaining the request of LLMs, we propose utilizing $h_R$ to perform multimodal information interaction with fine-grained image features. To this end, we adopt a multi-layer transformer blocks to achieve request-based visual information interaction. Each block is calculated as given in Eq. 1:

$$\mathbf{h}_l^R = \mathbf{LayerNorm}(\mathbf{h}_{l-1} + \mathbf{SelfAttention}(\mathbf{h}_{l-1})),$$
$$\mathbf{h}_l^F = \mathbf{LayerNorm}(\mathbf{h}_l^R + \mathbf{CrossAttention}(\mathbf{h}_I), \tag{1}$$
$$\mathbf{h}_l = \mathbf{LayerNorm}(\mathbf{h}_l^F + \mathbf{MLP}(\mathbf{h}_l^F)),$$

where $\mathbf{h}_{l\text{-}1}$ is the output of the $l$-1 th layer and the input of RVII is $h_R$. **SelfAttention** and **CrossAttention** are based on the multi-head attention mechanism, which are used to capture the desired visual information. After obtaining the output of last layer, we utilize a learnable linear project layer to transmit the interacted information to LLMs, which is denoted to $h_t$. Afterwards, the new presentation sequence ($h_{img}$, $h_X$, $h_t$) is fed into LLM to generate the final answer. Suppose that the training target of a multimodal instruction-following question answering is $Y = (y_1, ..., y_N)$, where $y_i$ represents the $i$-th token and $N$ refers to the total length, the optimizing loss is as following:

$$\mathcal{L} = -\frac{1}{N} \sum_{i=1}^{N} \mathbf{log} P_i(\hat{y}_i = y_i | h_{img}; h_X; h_{img\text{-}q}; y_1, ..., y_{i-1}). \tag{2}$$

## 3.2 Multimodal Instruction-following Tuning

We use various multimodal instruction-following data to makes interactive perception network effective. First, we construct two types of image-text semantic matching data based on the image-text pairs from datasets CC3M, COCO Caption, and Flick3k. The two types are "True or False" inference and four-choice selection tasks, respectively, where captions are randomly sampled from the corresponding training set. By doing so, the overall network could be trained to help and improve LLMs performing image-text semantic alignment. Second, to make LMEye adapt to various human queries, we introduce the multimodal instruction-following data about conversation and complex reasoning, released by Liu et al. (2023a). In addition, considering that a complex image contains infinite-level visual information and may be attached to external knowledge, we introduce data about the in-detail description of an image to improve multimodal long text generation capability. It includes the corresponding data from Liu et al. (2023a) and the artwork description dataset SemArt (Garcia & Vogiatzis, 2018). The total number of all instruction data is about 7.3M, encompassing 7.1M semantic

matching data, 20k artwork analysis samples, and 150k additional samples. Finally, identical to InstructBLIP, we also augmented the instruction-following data set by introducing partial training sets of approximately 20 multimodal tasks, while comparing them on two multimodal benchmarks.

# 4 EXPERIMENT

## 4.1 EXPERIMENTAL SETTINGS

**Datasets**. First, we evaluate LMEye on recently released comprehensive multimodal benchmarks: MMBench (Liu et al., 2023b) [1], and SEED-Bench (Li et al., 2023b), which are systematically-designed objective benchmark for robustly evaluating the various abilities of vision-language models. To verify the effectiveness of LMEye under various conditions, we also evaluate LMEye and other visual language models on three visual understanding and reasoning datasets: validation sets of VCR (Zellers et al., 2019) and VQAv2 (Goyal et al., 2017), and the test set of OK-VQA (Marino et al., 2019). In addition, we also use the GPT-3.5-turbo (OpenAI, 2023a) to produce five question-answer pairs centered around an image based on about 3.5k images and their long descriptions from Zhu et al. (2023). The prompt template is "*Generate five question-answer pairs for the following in-detail image description. Demand: the answer of question must be contained in the description and the type is Question: ... Answer: ... \n Description:* ". The total number of question-answer pairs is about 17.5k wherein the length of the answers exceeds that of conventional VQA datasets, with an average length of 13 words. The constructed data are used to evaluate and analyze performances of models.

**Comparing Models**. Flamingo (Alayrac et al., 2022) is a unifying multimodal generative model capable of rapidly adapting to a variety of image and video tasks. OFA (Wang et al., 2022a) is a sequence-to-sequence learning framework that could unify a diverse set of cross-modal and unimodal tasks. FROMAGe (Koh et al., 2023) is a typical LVLM that is efficiently trained by visually grounding LLMs with image captioning and contrastive learning, capable of image caption and image-text retrieval. BLIP-2 (Li et al., 2023c) is a two-stage training strategy that bootstraps vision-language representation learning and vision-to-language generative learning based on the frozen image encoder and language model, achieving state-of-the-art performances on various multimodal tasks. In addition, we also compare our methods with multimodal instruction-tuned model MiniGPT-4 (Zhu et al., 2023) and LLaVA (Liu et al., 2023a), where MiniGPT-4 is based on the pretrained Q-former from BLIP-2. Compared to BLIP-2 and FROMAGe, they are tuned with the multimodal instruction-following data generated by GPT-4. During the multimodal instruction tuning stage, both the projection matrix and LLM of LLaVA are updated.

**Implementation Details**. We run all experiments on eight Telsa A100-80G GPUs with the Python environment. To verify the effectiveness of LMEye, we adopt OPT-iml-1.3b, Bloomz-7b1, LLaMA-7b/13b (Touvron et al., 2023), and BLIP-2 (FlanT5$_{XL}$) as the backbone of our framework respectively. In feature alignment stage, we set the initial learning rate to 1e-4 and use the AdamW (Loshchilov & Hutter, 2017) optimizer to optimize the feature alignment process with the cosine declining way. The total training step of this stage is one epoch, and the batch size is 768. During the multimodal instruction tuning stage, we adopt a smaller batch size (256) and set the initial learning rate to 1e-4. The depth of RVII is set to 12 and the hidden size equals 768. We will freeze the first-stage parameters (include the linear project layer in feature alignment and the token representation of $$, or Q-former in BLIP-2) while performing multimodal instruction tuning. During generation, we employ the beam sample generation strategy from HuggingFace Transformer [2] repository and set the beam sizes to 4 and 1 for in-detail image description generation and VQA, respectively.

**Evaluation Metrics**. For visual question answering (VQA) with short answer and visual reasoning datasets, we adopt the common EM (exactly matching) calculation way as the evaluation method of accuracy. For the in-detail image description generation and VQA with long answer, we employ several generative evaluation metrics: BLEU (Papineni et al., 2002), ROUGE (Lin, 2004), CIDEr (Vedantam et al., 2015), and METEOR (Banerjee & Lavie, 2005).

---

[1]https://opencompass.org.cn/leaderboard-multimodal
[2]https://github.com/huggingface/transformers

Table 1: Model performances on MMBench. "TotalParams" indicates the total parameters of MLLMs. Logical Reasoning (LR), Attribute Reasoning (AR), Relation Reasoning (RR), Fine-grained Perception (Cross Instance) (FP-C), Fine-grained Perception (Single Instance) (FP-S), and Coarse Perception (CP).

| Models↓ Types → | TotalParams | Overall | LR | AR | RR | FP-S | FP-C | CP |
|---|---|---|---|---|---|---|---|---|
| OpenFlamingo (Alayrac et al., 2022) | 9B | 4.3 | 9.1 | 11.4 | 3.3 | 2.5 | 1.6 | 1.5 |
| OpenFlamingo v2 (Alayrac et al., 2022) | 9B | 5.7 | 11.4 | 12.8 | 1.4 | 5.5 | 0.8 | 4.0 |
| MiniGPT-4 (Zhu et al., 2023) | 8B | 23.0 | 13.6 | 32.9 | 8.9 | 28.8 | 11.2 | 28.3 |
| MMGPT (Gong et al., 2023) | 9B | 16.0 | 1.1 | 23.8 | 20.7 | 18.3 | 5.2 | 18.3 |
| PandaGPT (Su et al., 2023) | 14B | 30.6 | 15.3 | 41.5 | 22.0 | 20.3 | 20.4 | 47.9 |
| VisualGLM (Du et al., 2022) | 8B | 33.5 | 11.4 | 48.8 | 27.7 | 35.8 | 17.6 | 41.5 |
| InstructBLIP (Liu et al., 2023a) | 8B | 33.9 | 21.6 | 47.4 | 22.5 | 33.0 | 24.4 | 41.1 |
| LLaVA (Liu et al., 2023a) | 7.2B | 36.2 | 15.9 | 53.6 | 28.6 | 41.8 | 20.0 | 40.4 |
| LLaMA-Adapter-v2 (Gao et al., 2023) | 7.2B | 38.9 | 7.4 | 45.3 | 19.2 | 45.0 | 32.0 | 54.0 |
| G2PT | 7B | 39.8 | 14.8 | 46.7 | 31.5 | 41.8 | 34.4 | 49.8 |
| mPLUG-Owl (Ye et al., 2023) | 7.2B | 56.7 | 30.7 | 65.7 | 52.1 | 61.0 | 45.6 | 65.1 |
| Otter-I (Li et al., 2023a) | 9B | 48.3 | 22.2 | 63.3 | 39.4 | 46.8 | 36.4 | 60.6 |
| Shikra (Chen et al., 2023) | 7.2B | 60.2 | 33.5 | 69.6 | 53.1 | **61.8** | 50.4 | **71.7** |
| LMEye (ours) | **4.4B** | 62.3 | **40.3** | **74.7** | **55.4** | 61.7 | **58.0** | 68.9 |

Table 2: Model performances on SEED-Bench. We evaluate LMEye (FlanT5-XL)-4.4B on 9 dimensions for image understanding, including Scene Understanding (SU), Instance Identity (II), Instance Location (IL), Instance Attribute (IA), Instance Counting (IC), Spatial Relation (SR), Instance Interaction (IIR), Visual Reasoning (VR), and Text Recognition (TR).

| Models↓ Types → | Overall | SU | II | IL | IA | IC | SR | IIR | VR | TR |
|---|---|---|---|---|---|---|---|---|---|---|
| OpenFlamingo v2 (Alayrac et al., 2022) | 34.51 | 43.86 | 38.12 | 31.28 | 30.06 | 27.30 | 30.59 | 29.90 | 50.15 | 20.00 |
| MiniGPT-4 (Zhu et al., 2023) | 47.40 | 56.27 | 49.15 | 45.82 | 37.93 | 45.32 | 32.57 | 47.42 | 57.10 | 11.76 |
| MMGPT (Gong et al., 2023) | 34.54 | 43.64 | 37.85 | 31.45 | 30.78 | 27.34 | 30.14 | 29.90 | 51.36 | 18.82 |
| BLIP-2 FlanT5-XL (Li et al., 2023c) | 49.74 | 59.12 | 53.90 | 49.19 | 42.33 | 43.15 | 36.68 | 55.67 | 45.62 | 25.88 |
| InstructBLIP FlanT5-XL (Liu et al., 2023a) | 57.80 | 60.29 | 58.49 | 63.37 | 40.59 | 58.44 | 38.66 | 51.55 | 45.92 | 25.88 |
| InstructBLIP Vicuna (Liu et al., 2023a) | 58.76 | 60.20 | 58.93 | **65.63** | 43.56 | **57.05** | 40.33 | 52.58 | 47.73 | **43.53** |
| LLaVA (Liu et al., 2023a) | 36.96 | 42.69 | 34.90 | 33.45 | 28.43 | 41.85 | 30.75 | 27.84 | 46.83 | 37.65 |
| LLaMA-Adapter-v2 (Gao et al., 2023) | 35.19 | 45.22 | 38.50 | 29.30 | 33.03 | 29.67 | 35.46 | 39.18 | 51.96 | 24.71 |
| GVT (Wang et al., 2023) | 35.49 | 41.74 | 35.50 | 31.79 | 29.45 | 36.17 | 31.96 | 31.96 | 51.06 | 27.06 |
| VPGTrans (Zhang et al., 2023a) | 41.81 | 51.87 | 44.13 | 39.90 | 36.09 | 33.71 | 36.38 | 31.96 | 53.17 | 30.59 |
| mPLUG-Owl (Ye et al., 2023) | 37.88 | 49.68 | 45.33 | 32.52 | 36.71 | 27.26 | 32.72 | 44.33 | 54.68 | 18.82 |
| Otter (Li et al., 2023a) | 35.16 | 44.90 | 38.56 | 32.24 | 30.88 | 26.28 | 31.81 | 31.96 | 51.36 | 31.76 |
| LMEye (ours) | **59.70** | **73.20** | **64.12** | 56.57 | **53.99** | 48.75 | **47.64** | **65.98** | **76.13** | 37.65 |

## 4.2 OVERALL RESULTS

**MMBench Evaluation**. The evaluation results on the MMBench are presenteded in Table 1. The results show that our proposed model, LMEye-FlanT5-XL, outperformed other comparable models while using fewer parameters (4.4B vs. >7B). It is worth noting that LMEye outperformed other models in terms of reasoning performance, particularly in Logical Reasoning (LR), Attribute Reasoning (AR), and Relation Reasoning (RR). These indicate that LMEye is capable of effectively reasoning and making connections between different pieces of information, leading to better performance compared to other models. Additionally, InstructBLIP could be seen as an multimodal instruction enhanced LMEye variant without RVII, achieving interaction between human queries and image in Q-former, yet LMEye still outperforms it on multiple aspects.

**SEED-Bench Evaluation**. The experimental results presented in Table 2 demonstrate the effectiveness of LMEye in achieving state-of-the-art (SOTA) performance. Specifically, LMEye has shown significant improvements in scene understanding, with an increase of 13 points compared to previous SOTA. Moreover, in the category of sample attribute recognition and spatial connection understanding, LMEye also outperformed InstructBLIP. These results highlight the effectiveness of a plug-and-play interactive perception framework, in enhancing the ability of language models to understand images and multimodal instructions. Overall, these findings demonstrate the potential of LLMs in advancing the field of image understanding and suggest that plug-and-play interactive

Table 3: Zero-shot performances on some common multimodal datasets. LMEye variants with "*" indicate that we only remain the pretrained linear projection layer and remove the interactive process (RVII). "NumImg" represents the total number of images contained in feature alignment stage.

| Models↓ Types → | NumImg | VCR (Q→A) | VQAv2 | OK-VQA |
|---|---|---|---|---|
| Flamingo-3B (Alayrac et al., 2022) | >1B | - | 49.2 | 41.2 |
| MiniGPT-4 (Vicuna-7b) (Zhu et al., 2023) | 129M | - | 44.31 | 32.16 |
| LLaVA (Vicuna-7b) (Liu et al., 2023a) | 0.6M | - | 56.59 | 50.42 |
| OFA-Large (Wang et al., 2022a) | 20M | 25.07 | 40.25 | 19.34 |
| FROMAGe (OPT-6.7b) (Koh et al., 2023) | 3.3M | 20.87 | 44.08 | 20.06 |
| BLIP-2 (FlanT5$_{XL}$) (Li et al., 2023c) | 129M | 57.30 | 59.95 | 48.91 |
| InstructBLIP (FlanT5$_{XL}$) (Liu et al., 2023a) | 129M | 53.55 | 62.56 | 50.12 |
| LMEye (OPT-iml-1.3b)* | 1.7M | 34.34 | 38.34 | 22.26 |
| LMEye (OPT-iml-1.3b) | 1.7M | 39.52 | 42.42 | 24.58 |
| LMEye (Bloomz-7b1)* | 13M | 39.31 | 40.63 | 25.56 |
| LMEye (Bloomz-7b1) | 13M | 43.07 | 42.20 | 26.38 |
| LMEye (Bloomz-7b1)* | 69M | 42.81 | 42.39 | 26.79 |
| LMEye (Bloomz-7b1) | 69M | 47.00 | 45.58 | 35.11 |
| LMEye (BLIP-2-FlanT5$_{XL}$) | 129M | **57.50** | **63.20** | **54.0** |

Table 4: Ablation experiments on the self-constructed evaluation benchmark.

| Models↓ Types → | NumImg | VQA with Long Answer | | | | | |
|---|---|---|---|---|---|---|---|
| | | B@1 | B@2 | R@1 | R@L | CIDEr | METEOR |
| LLaVA (Vicuna-7b) | 0.6M | 29.23 | 17.62 | 40.85 | 38.67 | 166.95 | 50.15 |
| LMEye (Bloomz-7b1)* | 13M | 2.15 | 0.40 | 7.32 | 7.24 | 7.49 | 4.47 |
| LMEye (Bloomz-7b1) | 13M | 37.49 | 23.31 | 34.94 | 33.41 | 131.83 | 46.61 |
| LMEye (Bloomz-7b1)* | 69M | 3.08 | 0.52 | 7.08 | 7.0 | 7.51 | 4.67 |
| LMEye (Bloomz-7b1) | 69M | 34.77 | 22.0 | 39.20 | 37.37 | 167.17 | 48.60 |
| LMEye (LLaMA-7b) | 1.7M | 42.40 | 27.84 | 41.06 | 39.37 | 188.59 | 50.22 |
| LMEye (LLaMA-13b) | 1.7M | **49.36** | **33.67** | 43.96 | 42.05 | **212.16** | **51.24** |
| BLIP-2 (FlanT5$_{XL}$) | 129M | 10.60 | 5.43 | 25.29 | 24.21 | 61.88 | 17.75 |
| LMEye (BLIP-2-FlanT5$_{XL}$) | 129M | 41.56 | 26.49 | **47.14** | **44.46** | 193.92 | 50.63 |
| Models↓ Types → | NumImg | In-detail Image Description | | | | | |
| | | B@1 | B@2 | R@1 | R@L | CIDEr | METEOR |
| LMEye (Bloomz-7b1)* | 13M | 10.55 | 1.73 | 5.78 | 5.60 | 0.8 | 7.02 |
| LMEye (Bloomz-7b1) | 13M | 27.50 | 6.0 | 12.0 | 11.68 | 2.4 | 21.15 |
| LMEye (Bloomz-7b1)* | 69M | 11.59 | 2.37 | 6.92 | 5.68 | 0.6 | 11.15 |
| LMEye (Bloomz-7b1) | 69M | 26.27 | 6.71 | 14.15 | 13.80 | 2.5 | **26.71** |
| LMEye (LLaMA-7b) | 1.7M | 26.21 | 6.47 | 11.18 | 10.86 | 1.4 | 26.02 |
| LMEye (LLaMA-13b) | 1.7M | **29.91** | **7.11** | 12.06 | 11.79 | **3.8** | 23.06 |
| BLIP-2 (FlanT5$_{XL}$) | 129M | 1.21 | 0.56 | **18.37** | **17.48** | 2.3 | 8.15 |
| LMEye (BLIP-2-FlanT5$_{XL}$) | 129M | 8.98 | 2.47 | 17.06 | 16.17 | 3.0 | 14.05 |

perception frameworks can be an effective means of leveraging these capabilities. Further research in this area may pave the way for more sophisticated and effective approaches to image understanding, with implications for a wide range of applications and industries.

## 4.3 ABLATION STUDY AND ANALYSIS

**Visual Question Answering and Multimodal Reasoning**. The experimental results are presented in Table 3, where we do not present the VCR result of LLaVA and MiniGPT-4 since they do not follow the prompt to select one option from four candidates. This may be attributed to the self-ability of Vicuna Chiang et al. (2023), which is the LLaMA tuned only with conversation data. Compared to conventional VLMs such as Flamingo-3B and OFA, the designed LMEye variants and other MLLMs can be able to achieve better zero-shot performance on answer selection (VCR) and short answer generation tasks (VQA), even in the case that LMEye (Bloomz-7b1) have only seen 1.7M images

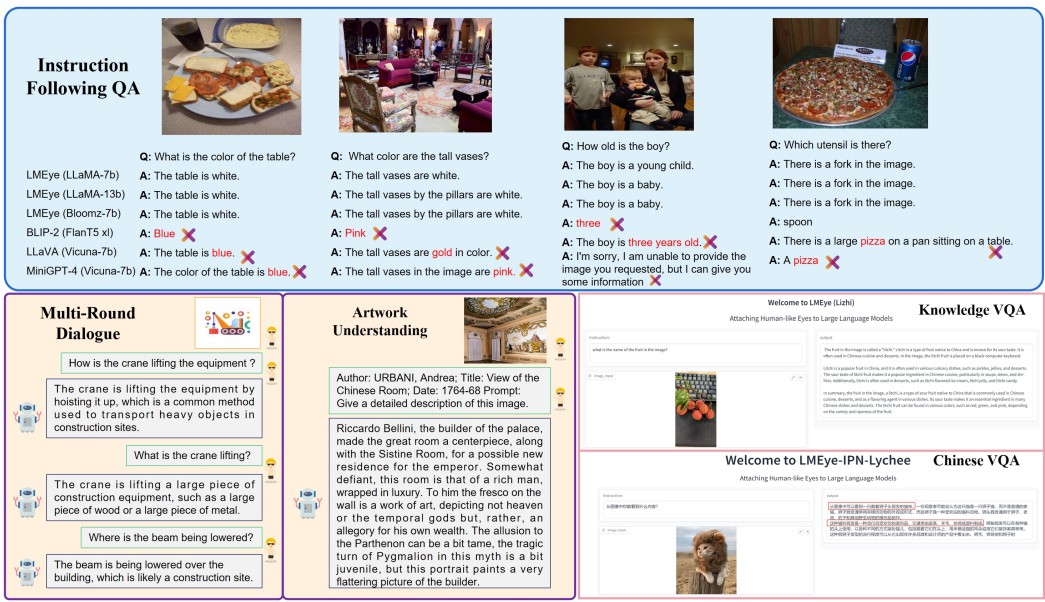

Figure 2: Illustration of some cases generated by various LMEye variants. Artwork Understanding and Multi-Round Dialogue are based on the LMEye (FlanT5-XL). Knowledge and Chinese VQA are based on a Chinese-English bilingual language model: Lychee-Base-11B. IPN represents the interactive perception network and Lizhi is the Chinese pinyin of Lychee.

during the pretraining stage. Hence, it is feasible to adopt such a training efficient MLLM construction method based on the frozen visual encoder and LLM. In addition, introducing more powerful language models and high-quality image-text data in the pretraining stage will improve the accuracy of the language model in understanding image information, e.g., the performance comparisons of various LMEye (Bloomz-7b1) and LMEye (OPT-iml-1.3b) variants. When we introduce LMEye based on BLIP-2 and train the interactive framework through collected multimodal instruction data, it substantially improves performance on the complex visual problem task OK-VQA by about 5% and could achieve better performance than InstructBLIP. By further comparing their performances in Tables 1 and 2, we can find that the effectiveness of introducing the two-stage interactive perception network (RVII).

**VQA with Long Answer and In-detail Description**. We mainly evaluate various LMEye variants on the constructed evaluation benchmark: in-detail image description and visual question-answering tasks. According to the experimental results shown in Table 4, we can see that multimodal instruction-tuned LMEye models significantly improve almost all generative metrics. Combined with the given examples shown in top part of Figure 2, we suggest that the multimodal instruction-following tuning approach is helpful for LLMs to achieve an image understanding capability similar to GPT-4. In addition, we find that LMEye (Bloomz-7b1) could understand the intent of various questions and generate an accurate response. Compared with LLaVA (Vicuna-7b), MiniGPT-4, and BLIP-2, which incorporate the static visual information for different questions about an image, our method can obtain corresponding visual information relevant to the human queries and generate more accurate responses (see comparative performances in experimental Tables and Figure 2).

Unlike the multimodal instruction-following data used by MiniGPT-4 and LLaVA, we introduce the artwork description data as part of instruction-tuning data and thus improve the model's capability to understand the artwork. We observe that LLMs could utilize their stored knowledge to present sufficient analysis for the artwork, as the Artwork Understanding shown in in Figure 2. From Table 4, we also observe that the improvement of detailed image description ability mainly comes from using relevant instruction data. Our method mainly improves the performance of MLLMs on VQA tasks with various queries. We present more cases in Appendix. In conclusion, the ablation experiments of LMEye variants show that the proposed interactive perception network could be play-and-plug in

various large language models and improve the overall performance by incorporating request-based interacted visual information module.

## 5 DISCUSSION AND FUTURE WORK

**Instruction-tuned LLMs can better generalize on multimodal tasks**. Previous work (Li et al., 2023c) shows that the BLIP-2 variant with instruction-tuned LLMs performs best on many multimodal tasks. In Table 3, we observe that LMEye (OPT-iml-1.3b)* is capable of better performance on VCR and OK-VQA compared to FROMAGe (OPT-6.7b) with a larger-scale OPT version. This could be attributed to the fact that text-only instruction-tuned LLMs better understand human input queries than original LLMs. Thus they have better performances on multimodal QA tasks.

**Quality and Diversity of multimodal instruction-following data are important**. A comparison between LLaVA (Vicuna-7b) and MiniGPT-4 (Vicuna-7b) reveals that LLaVA, which incorporates a larger number of diverse multimodal instruction data, outperforms MiniGPT-4. This finding is consistent with the research conducted by Liu et al. (2023a), who demonstrate that diverse multimodal instruction data can enhance the overall performance of MLLMs across various tasks. Current multimodal instruction-following data are usually constructed by powerful GPT-4 through the Self-Instruct (Wang et al., 2022c) technical. While these automatically generated instruction data exhibit diversity, there remains room for improvement in terms of quality. In the future, it would be advantageous to incorporate high-quality multimodal task data, including video, image, and audio, to enhance the comprehensive capability of instruction-tuned MLLMs.

**Visual information should interact with human instruction**. Previous work InstructBLIP attempts to input textual questions into the Q-former to refine its performance in the specific visual question-answering task, leading to superior results. These questions facilitate visual information extraction by utilizing self-attention layers within the Q-Former architecture. Different from BLIP-2, LMEye focuses on extracting image features that are highly informative for the encoding request from LLMs, achieving dynamic interaction between LLMs and visual information. Additionally, we introduce diverse multimodal instruction-following data to train LMEye, allowing them to adapt to a wide range of human queries. Consequently, LLMs can leverage enriched visual information to accomplish different tasks effectively. To conclude, enabling visual information to interact with human instruction is effective for improving the capability of MLLMs.

**Hallucination**. While MLLMs generate in-detail image description or artwork analysis, they easily produce fragments that are nonsensical or unfaithful to the objective image and common knowledge, or fabrication of facts. Some cases are shown in Appendix. To address this issue, in the future, we can introduce the alignment technical (such as Reinforcement Learning from Human Feedback (RLHF) (Ouyang et al., 2022)), retrieval augmentation, or multimodal chain-of-the-thought (COT) (Wei et al., 2022) to improve the factuality of generated content.

## 6 LIMITATION

While our models strive to enhance their alignment with human queries, it is important to acknowledge that they are not completely aligned nor entirely safe. Despite efforts to improve the quality of outputs, our models still have limitations in avoiding generating toxic or biased content, fabrication of facts, and other undesirable outputs. In some cases, the model may inadvertently generate offensive, discriminatory, or harmful outputs. It can be attributed to biases in the training data or the self-ability of LLMs. Furthermore, due to constraints in the quality and diversity of available multimodal instruction-following data, the model may provide incorrect responses for certain queries.

## 7 CONCLUSION

We present LMEye, attaching human-like eye for large language models with an interactive perception network, aiming to achieve a large visual language model via dynamic interaction between LLMs and visual information. The experimental results show that our method with less parameters achieves superior performances on two evaluation benchmarks of multimodal LLMs, and other visual question answering, in-detail image description, and multimodal reasoning tasks.

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
