## A  VISUAL QUESTION ANSWERING

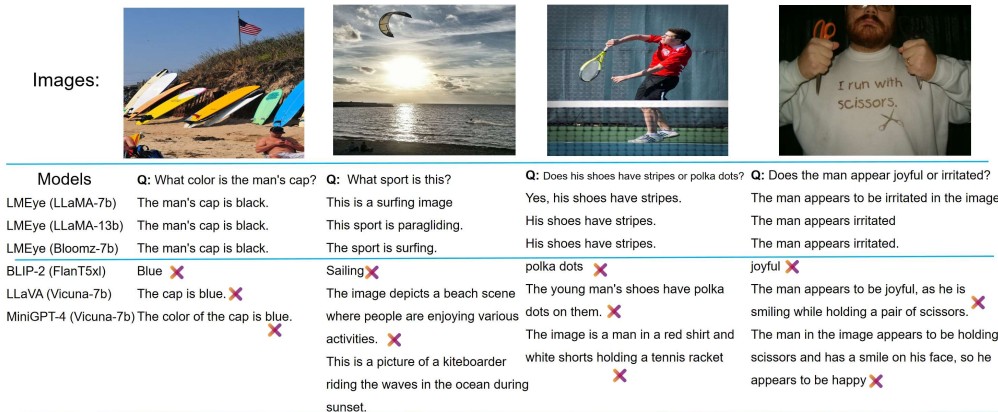

Figure 3: Illustration of VQA cases generated by several models.

## B  MULTI-TURN DIALOGUE FOR ONE IMAGE

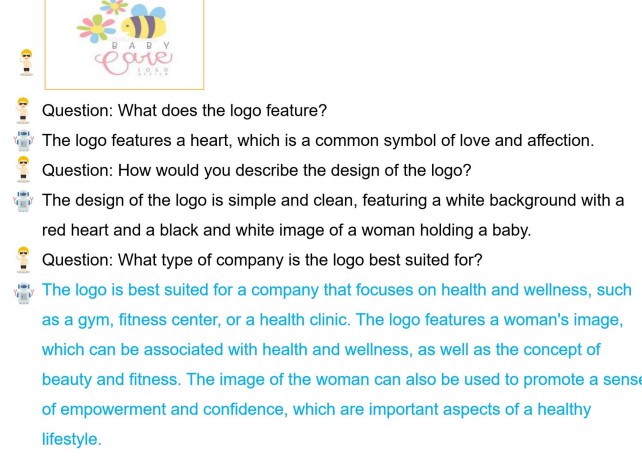

Figure 4: A multi-turn question answering case generated by LMEye (LLaMA-7b).

## C  IN-DETAIL IMAGE DESCRIPTION

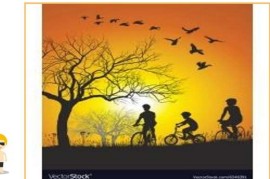

Question: What does the image depict?

The image depicts a beautiful sunset scene with a mountain in the background. The sun is setting behind the mountain, creating a stunning view.

Question: How does the tree and birds in the background add to the image?

The tree and birds in the background add to the image by providing a sense of nature and a connection to the outdoors. The tree provides a sense of stability and a natural setting, while the birds add a sense of life and movement to the scene. The combination of these elements creates a peaceful and serene atmosphere, which enhances the overall aesthetic of the image.

Figure 5: Another multi-turn question answering case generated by LMEye (LLaMA-7b).

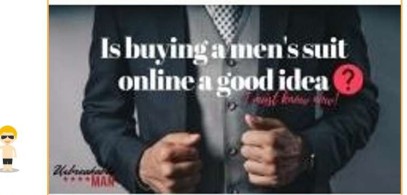

Question: What is the man wearing in the image?

The man is wearing a suit and tie.

Question: What is the color scheme of the image?

The color scheme of the image is black and white.

Question: What is the man's expression in the image?

The man's expression in the image is serious.

Question: What could the image be used for?

The image can be used for a variety of purposes, such as advertising, marketing, or promotional materials. It can also be used for educational or informational purposes, such as in a textbook or presentation.

Question: Is the man wearing gloves in the image?

Yes, the man is wearing gloves in the image.

Figure 6: A multi-turn question answering case generated by LMEye (LLaMA-13b).

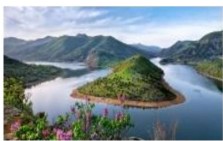

BLIP-2: a lake surrounded by mountains and flowers

LLaVA (Vicuna-7b): The image features a serene lake surrounded by mountains and hills, with a large island in the middle of the lake. The lake is calm and blue, reflecting the beautiful landscape. There are several small boats scattered across the lake, some near the shore and others further away. In the background, there are mountains of various sizes, adding to the picturesque scenery. The scene is a perfect representation of a peaceful, natural environment that one can find in the mountains.

LMEye (LLaMA-7b)The image captures a beautiful scene of a lush green forest with tall trees and a river flowing through the area. The river runs through the forest, creating a peaceful and tranquil atmosphere. There are several trees in the scene, some of which are positioned closer to the river, while others are further away from the water. The tall trees create a dense canopy, providing shade and enhancing the natural beauty of the area. In addition to the trees and the river, there are a few rocks scattered throughout the forest.These rocks add to the natural beauty of the scene, creating a picturesque landscape. The overall atmosphere of the scene evokes a sense of serenity and tranquility, making it a perfect place to relax and unwind.

LMEye (LLaMA-13b): The image shows a breathtaking view of a mountainous landscape with a river flowing through it. The landscape is filled with greenery, creating a lush and vibrant scene. The river appears to be winding its way through the mountains, adding to the scenic beauty of the area. The image captures the tranquility and serenity of the natural environment, inviting the viewer to take a moment to admire the stunning view.

LMEye (Bloomz-7b1): The image depicts a beautiful, serene scene with a large body of water, surrounded by lush green foliage. The water is surrounded by mountains, creating a dramatic backdrop for the scene. There is a small island in the middle of the body of water, giving it a sense of seclusion and isolation. In the middle of the body of water, there is a small boat with two people on board. They seem to be enjoying a peaceful day on the water, gazing at the beautiful scenery. The boat is positioned in the middle of the body of water, creating a sense of balance and harmony in the scene. (Hallucination)

Figure 7: An in-detail description case generated by several models.

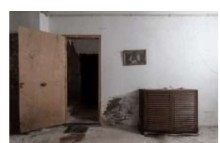

BLIP-2: an empty room with a fan in it

LLaVA (Vicuna-7b): The image features a large, dark room with a doorway that has a white frame. The doorway is open, leading to a hallway that appears to be empty. Inside the room, there is a brown, rusted metal cabinet that stands out in the space. In addition to the cabinet, there are several books scattered throughout the room, adding to the overall cluttered appearance. The books can be seen on the floor, as well as on various surfaces within the room. The room seems to be in a state of disarray, with the open doorway leading to the equally mysterious hallway beyond.(Hallucination)

LMEye (LLaMA-13b): The image depicts a room with a white wall and a door on the left side. The room appears to be empty, with no furniture or people present. The door appears to be closed, suggesting that the room is not currently in use.

LMEye (LLaMA-7b):The image depicts a room with a large wooden desk placed in the center of the space. The desk is covered with various items, including a clock, a book, a pencil, and a calculator. There is also a vase placed on the desk, adding a touch of elegance to the room. In addition to the desk, there are two chairs placed in the room, one on each side of the desk. The room appears to be a classroom or an office, as there are several books placed on the desk and around the room.(Hallucination)

LMEye (Bloomz-7b1): The image depicts a small room filled with various pieces of furniture, including a table, a chair, and a bed. The furniture appears to be in good condition, with the table, chair, and bed all in good condition. The room is filled with a variety of items, including a bookshelf, a lamp, and a mirror. The bookshelf is placed in the corner of the room, while the lamp and mirror are positioned in the middle of the room. There is also a small painting on the wall of the room, which adds a touch of color to the scene. Overall, the room appears to be in good condition and ready for use.(Hallucination)

Figure 8: Another case of generated in-detail description.