# OpenReview forum: "LMEye: An Interactive Perception Network for Large Language Models"
_ICLR.cc/2024/Conference — Submitted to ICLR 2024_

### Official Review · Reviewer_RpWw · 2023-10-31

**Soundness:** 4 excellent
**Presentation:** 3 good
**Contribution:** 4 excellent
**Rating:** 8
**Confidence:** 5

**Summary:**

Recognizing the challenges of training Multimodal Large Language Models (MLLM) from scratch, the authors propose LMEye, an interactive perception network that acts as a "human-like eye" for LLMs. In contrast to previous methods, LMEye facilitates a dynamic interaction between the LLM and external visual information, akin to Visual LLM Agent . This is accomplished by enabling the LLM to specify the kind of visual information it needs based on the human instructions it receives. The LMEye system is composed of a basic visual mapping network for initial image perception and novel RVII modules that can receive visual information requests from the LLM, process these requests, and subsequently relay the interacted visual information back to the LLM. This method ensures that the LLM not only receives visual data but can actively seek out specific visual information in alignment with human instructions. Experimental evaluations on various multimodal benchmarks reveal that LMEye notably enhances zero-shot performance on diverse multimodal tasks (MM-Bench and SEED-Bench), suppressing prior methods, and doing so with fewer parameters (LMEye-4B vs. MLLMs >7B).

**Strengths:**

1. Overall Novelty and Contributions:
The LMEye approach stands out as a monumental stride in the integration of visual perception with LLMs. The unique design of Request-based Visual Information Request (RVII), allowing LLMs to 'request' specific visual information, is both intuitive and groundbreaking, which reflects a deep understanding of how multimodal processing might emulate human brain-like cognition. The motivation is clear and interesting, and the experiments are solid for supporting their ideas. The impressive results on two comprehensive multimodal benchmarks validate the effectiveness of LMEye (less parameters, better results). The section of Discussion and Future Work present the current development of Multimodal Large Language Models and give useful advice for constructing a multimodal language model, such as selecting language models and how to design interaction between human instructions and visual information. This paper might become a foundational reference for future studies about LLMs as the core processor of Multimodal Agent.

2. Presentation:
The overall description of the paper is clear and the motivation of incorporating human-like eyes for LLMs is novel, different from construction of current multi-modal large model. The extensive experiments including two widely-used benchmarks, other downstream datasets, and self-constructed benchmarks, fully verified the author's motivation and proposed approach. The experimental analysis are solid.

3. Method:
a. LMEye introduces dynamic visual information interaction between LLMs and objective Visual Information. By recognizing and addressing the gap between static and dynamic visual information processing, LMEye offers a more holistic approach to multimodal understanding. It allows LLMs to actively request and incorporate relevant visual content based on specific human instructions, ensuring a more tailored and context-aware response. By preserving the inherent structure of LLMs, LMEye ensures that their original performance on NLP tasks is not compromised, thereby maintaining the robust generalization abilities of LLMs.
b. A novel Request-based Visual Information Interaction module (RVII) is introduced. This module enables the LLM to understand the basic visual information and human queries, send a request for additional specific visual information, and then construct a response based on the integrated understanding of the image, text instruction, and the interacted visual information. As we know, when humans complete tasks according to instructions, they usually interact with the external environment multiple times.
c. The whole training process is parameter-efficient and RVII module could be injected into various LLMs, e.g., Llama, Bloom, BLIP-2, and FlanT5-xl. The scalability of the proposed method, LMEye: interactive perception network, is easy and stable.

4. Experiments:
a. Superior Performance with Fewer Parameters: The results from the experiments are commendable. LMEye demonstrates superior performance in multimodal understanding and reasoning when tested on two comprehensive evaluation benchmarks: MMBench and SEED-Bench. What's striking is its ability to outperform other MLLMs while using substantially fewer parameters (4.4B for LMEye vs. >7B for others). This suggests that LMEye isn't an incremental improvement but offers a significant leap in the direction of efficient and effective multimodal modeling.
b. Consistent Improvements in Ablation Studies: Ablation studies further solidify the effectiveness of LMEye. There is a significant improvement in zero-shot multimodal performances across different scales and types of LLMs. The exact gains mentioned, such as a 5.0% improvement on OK-VQA for LMEye (BLIP-2) in comparison to BLIP-2 alone, and a remarkable 20% gain on VQA with long answer for LMEye (LLaMA-7b) against LLaVA (Vicuna-7b), speak volumes about the robustness of the method. This consistency in improvement across different benchmarks and tasks emphasizes the versatility and generalizability of LMEye.
c. The authors present intriguing cases that highlight several noteworthy aspects of their proposed method. These include multi-round interactions between Large Language Models (LLMs) and visual information, showcasing the model's ability to handle multilingual data, as well as its capability for artwork analysis. These examples provide valuable insights into the versatility and potential applications of the proposed approach.
d. The comprehensive Section: Discussion and Future Work in the paper serves as a valuable synthesis of the key steps involved in constructing large multimodal models. It substantiates the claims made by the authors with corresponding experimental evidence.

**Weaknesses:**

1. The experimental part should offer a concise description of the structure of LMEye (BLIP-2).
2. The caption and top line of tables 1 and 2 should be spaced wider.
3. In the section discussing multimodal instruction-following part, it would be beneficial to provide a more comprehensive description of data, such as a figure. Specifically, elucidating the various categories of images within the dataset would offer readers a clearer understanding of its composition and scope.
4. Whether the inference process will consume some time for different types of models, a brief inference efficiency description is given for the design of the LMEye variant for the encoder-decoder and the decoder-only models.

**Questions:**

1. See 3 and 4 in Weakness.
2. Would this approach expand to video understanding and provide more detailed description of video and action inference?
3. To alleviate the Hallucination problem shown in Appendix, could we introduce visual information request during text generation, like ReAct?

---

> ### Author Response · Authors · 2023-11-15
> **Response to the constructive comments from Reviewer RpWw**
>
> Thank you for your appreciation and acknowledgment of our work. Below, we will address your constructive comments individually.
>
> **Q1**: Give a short description of LMEye (BLIP-2-FlanT5-XL) and provide more clear data introduction.
>
> **A**: LMEye (BLIP-2-FlanT5-XL) is a derivative of the FlanT5 large language model, operating on an encoder-decoder architecture. It employs the pre-trained vision encoder along with the Q-former to serve as the image encoder, translating image representations into the language space. Specifically, we utilize the T5-Encoder for image comprehension and request processing, employing a probe vector labeled as '<img-q>' to prompt signals essential for the large model's visual information acquisition. Subsequently, we leverage the RVII mechanism to gather visual information relevant to the aforementioned model requests. This process enriches the original image data with dynamically enhanced visual information. The amalgamation of this enhanced visual data and the original image information is then fed into the T5 model.
>
> In terms of multimodal instruction data, our approach encompasses datasets akin to those utilized by InstructBLIP, such as VQA and OK-VQA. However, our adaptation goes further by integrating additional sources beyond the initial scope, including artwork analysis and LLaVA data.
>
> **Q2**: The difference between the LMEye variants for the encoder-decoder and the decoder-only language models.
>
> **A**:
> The distinction between the LMEye variants in encoder-decoder and decoder-only language models lies in their operational mechanisms. Encoder-decoder LLMs leverage the encoder specifically to grasp input information, enhancing model inference efficiency by focusing on relevant data. On the other hand, decoder-only LLMs utilize the entire language model to accommodate evolving visual input and human queries. However, our recent advancements have led to an enhanced variant that integrates the foundational layer of decoder-only LLMs for comprehending both image and query data. Moreover, we've implemented the RVII (small size) module across the upper layers of the language model. This strategic integration empowers the language model to continually assimilate dynamically enriched visual information as it delves deeper, akin to imbuing deep thinking with a visual dimension. Consequently, this visual data becomes an integral component of the calculation process within LLMs, augmenting their cognitive capabilities.
>
> **Q3**: Expand LMEye to Video understanding.
>
> **A**:
> This question delves into a fascinating area of exploration. Extensive experimentation has convincingly showcased the remarkable scalability of LMEye across various large language models, hinting at its potential extension into the realm of video processing. The approach involves training a dedicated video encoder and converter, leveraging RVII's ability to engage with prolonged encoded signals. Our methodology encompasses aggregating encoding information from brief video frames and subjecting it to rigorous benchmark testing for video action and procedural comprehension.
> | Models             | Action Prediction | Procedure Understanding |
> |--------------------|-------------------|-------------------------|
> | VPGTrans           | 39.54             | 31.88                   |
> | LLaMA-Adapter V2   | 38.56             | 19.63                   |
> | MiniGPT-4          | 38.22             | 27.1                    |
> | InstructBLIP Vicuna| 34.48             | 23.07                   |
> | InstructBLIP       | 33.10             | 27.1                    |
> | BLIP2              | 32.59             | 23.99                   |
> | LLaVa              | 29.71             | 19.13                   |
> | mPLUG-Owl          | 26.72             | 26.51                   |
> | Video-ChatGPT      | 27.59             | 21.14                   |
> | LMEye(ours)        | **46.55**             | **35.15**                   |
>
> The outcomes of our experiments substantiate the feasibility of extending LMEye's capabilities to comprehend video content. Notably, our findings highlight not only the successful extension of this model but also its commendable performance compared to Video-related MLMs. This progress indicates promising avenues for further advancements and applications in this evolving field.
>
> **Q4**: Introduce visual information request during text generation
>
> **A**:
> We are actively considering the integration of visual information retrieval mechanisms into our text generation process. Our current focus is on the development of specific tokens dedicated to retrieving pertinent visual data during generation. This approach mirrors our successful implementation of Tool API tokens and aligns with our ongoing efforts to enrich the generation process with relevant external information sources. Your feedback has guided us towards this promising avenue, and we are committed to exploring and implementing this enhancement in our future work.

---

### Official Review · Reviewer_DT9V · 2023-10-31

**Soundness:** 2 fair
**Presentation:** 2 fair
**Contribution:** 2 fair
**Rating:** 5
**Confidence:** 4

**Summary:**

This paper proposes a multimodal LLM, LMEye. Different from preivous works using a simple mapping network to coneect vision and LLM, the visual information to LLM of LMEye is language-query conditioned. It first utilizes a LLM to extract the information of both vision and language, followed by a Request-based Visual InformationInteraction module to fuse vision and language signal. Finally the fused information and human queries are fed into another LLM to complete the instruction. The proposed method is evaluated on comprehensive multimodal benchmark like MMBench and SEED-Bench, traditional tasks like VQAv2 and a self-collected QA dataset, where LMEye achieves promising results.

**Strengths:**

- The motivation and implementation are clear and straightforward. The visual information should interact with language queries in multimodal LLMs. Current MLLMs does not point out this issue but I think it is important for deeper understanding of the visual signal.
- The evaluation on multiple tasks are promising. I believe LMEye can serve as a strong baseline for future works.
- A new VQA dataset with long answer. Traditional VQA datasets are not suitable for current MLLMs since their answers are usually one word or phase, while MLLMs are tend to generate long and detailed answers. So this dataset will motivate the research community to pay more attention to improve MLLM's problem solving capability rather than fitting to traditional VQA benchmarks.

**Weaknesses:**

- As the core contribution of the paper, the authors did not carefully explore the contribution of RVII module. For example, the impact of  RVII module under the same data and training process.
- In my view, I think there is no essential difference between RVII and LoRA. LoRA is too insert some parameters inside the LLM , while RVII is more like an Adapter module outside LLM. The authors claim that visual feature from RVII is dynamic and conditioned on human queries. Since the decoding process of LLM are autogressive, a token to be decoded is also conditioned on privous visual tokens and human queries. The LoRA layer will take as input both visual tokens and human queries.

**Questions:**

- What are the training parameters in the instruction tuning phase?
- Since the visual information needs to be fed into LLM twice, will the model run slower?

---

> ### Author Response · Authors · 2023-11-15
> **Respond to the valuable comments of Reviewer DT9V**
>
> Thank you for your suggestions and comments on our work. We will answer your questions one by one in two pages.
>
> **Q1**: Explore the contribution of RVII module. For example, the impact of RVII module under the same data and training process.
>
> **A**: The RVII module is designed to enhance the model's ability to dynamically interact with visual information based on human queries. To evaluate its contribution, we refer to Tables 3 and 4, where we compare the performance of models with and without the RVII module (notably LMEye (xx)*). These tables show that incorporating RVII results in significant improvements for several LLMs in tasks like VQA/OK-VQA/VQA with long answers. Moreover, when comparing LMEye (FlanT5-BLIP-2-XL) with Instructblip (FlanT5-BLIP-2-XL), both of which utilize similar multimodal instruction data, the key difference lies in the inclusion of the RVII structure in our model. This comparison, as shown in the results on evaluation benchmarks, also underscores the effectiveness of the RVII module.
>
> To further explore the contribution of RVII module, we conducted an additional experiment. Here, we fine-tuned LMEye (FlanT5-BLIP-2-XL)* using identical instruction data but without the RVII module. The results of this experiment, detailed in the following table, clearly demonstrate the significant contribution of the RVII module to the model's performance.
> | Models(FlanT5-XL) | MMBench.Test | SEEDBench | VCR  | VQA  | OK-VQA |
> |-------------------|--------------|-----------|------|------|--------|
> | BLIP-2            | -            | 49.74     | 57.3 | 59.95| 48.91  |
> | InstructBLIP      | 33.9         | 57.8      | 53.55| 62.56| 50.12  |
> | LMEye(ours) w/o RVII | 47.79   | 54.84     | 53.51| 56.72| 49.6   |
> | LMEye(ours)       | **62.6**         | **59.7**      | **57.5** | **63.2** | **54**     |
>
> **Q2**: The training parameters in the instruction tuning phase?
>
> **A**: During the instruction fine-tuning stage, our model utilizes the RVII system along with several linear feature projection layers, which collectively amount to approximately 105 million parameters, with RVII alone comprising about 100 million. Recent findings indicate that for image understanding tasks, it's possible to reduce the number of RVII layers without noticeably affecting performance. By halving the number of RVII layers, we can significantly decrease the total count of trainable parameters, thereby streamlining the model while maintaining its effectiveness.
>
> **Q3**: Since the visual information needs to be fed into LLM twice, will the model run slower?
>
> **A**: Thank you for raising this point. Intuitively, feeding visual information into the LLM twice suggest a slower process. In practice, the impact on processing speed is minimal. This is due to the efficient design of our model which optimizes the integration of visual data. The length of visual tokens is set to 32 and we only input one <img-q> token as the probing token to obtain the request signal. For encoder-decoder T5 model, we only use the encoder as the the first-stage basic perception and request signal generation. Time is mainly spent in the token generation phase. Moreover, the additional time taken for processing is offset by the significant improvements in accuracy and context relevance of the output. In summary, while there is a slight increase in processing time, it is negligible compared to the enhanced performance and quality of results achieved by our model and we will contiguously optimize the efficiency according to your question.

---

> ### Author Response · Authors · 2023-11-15
> **Continue to respond to the final comment**
>
> **Q4**: In my view, I think there is no essential difference between RVII and LoRA.
>
> **A**: Thank you for your insightful question. To clarify the distinctions between RVII and LoRA, we focus on three aspects: integration approach, adaptation dynamics, and role in the decoding process.
>
> **Distinction in Integration Approach**
> Regarding LLMs as an agent, The RVII module is designed to obtain the dynamic visual information based on the interaction between LLMs and visual information.  LLMs first understand the basic image content and human queries in the first-stage perception process, then sending its request signal to obtain the visual information what they desire according to its understanding. The dynamic enhancing visual information are used in the second-stage thinking (response generation) process. LoRA integrates additional parameters within the LLM and alters the internal dynamics of the LLM by tweaking its parameters, used to focus on valuable visual and textual information in the internal operations of LLMs. The influential LLaMA-adapter V1/2 has made significant progress in this regard. In contrast, RVII acts as an external intermediary, processing visual inputs based on real-time human queries before they are explicitly fed into the LLM to generate final response. *RVII's design preserves the LLM's original structure, treating it as an agent, highlights the importance of external and adaptable multimodal integration and make up for the information loss caused by one-step visual input. Of course, we believe that using LoRA to further tune LLMs will enhance the performance of our model*.
>
> **Dynamic vs. Static Adaptation**
> Another key difference lies in the nature of adaptation. RVII dynamically interacts with visual data, adjusting its processing based on the context of human queries. This is in contrast to LoRA's adaptation, where the incorporated changes are fixed post-training and visual information do not dynamically alter based on the human query. RVII's dynamic adaptation allows for more context-sensitive and relevant integration of visual information, enhancing the LLM's response accuracy to specific human queries. Similarly, based on your suggestions, we can also learn from the recently proposed Image-Blind (we will introduce and compare it in the next version) and Llama-adapter work, introduce the second stage of training and use the LoRA adapter in large models to improve the operation of two types of visual information and textual query during decoding progress.
>
> **Role in Decoding Process**
> While it is accurate that in LLMs, each token is decoded based on previous tokens, including visual tokens and human queries, RVII adds a nuanced dimension to this process. RVII continuously refines the visual input, ensuring that at each step of the decoding, the visual information remains contextually aligned with the evolving user query. This ongoing adjustment keeps the visual data relevant throughout the interaction, providing a more sophisticated and tailored integration than static models. In summary, RVII's external, dynamic, and context-aware approach to integrating visual information with LLMs offers distinct advantages over one-step inputting static image feature in LLMs, particularly in terms of real-time adaptability and relevance to specific user queries. The LoRA mechanism plays a crucial role in refining the interaction between textual and visual inputs, ensuring a more seamless and effective multimodal integration. Hence, **we believe that the advantages of LoRA and RVII could be integrated together**. We plan to use LoRA technical to further improve the adaptation of LLMs on multimodal input and performance, yet we also need consider another question, that is remaining the original capability of LLMs to process text-only NLP tasks.

---

### Official Review · Reviewer_c2o8 · 2023-11-08

**Soundness:** 2 fair
**Presentation:** 2 fair
**Contribution:** 2 fair
**Rating:** 3
**Confidence:** 4

**Summary:**

This paper addresses the language-vision tasks by proposing a dynamic attetion-based model that uses human questions/instructions as input to dynamically query visual features for better visual information summarization. The authors claim the method is better than Blip model due to interactive attention and show promising results.

**Strengths:**

+ The overall motivation of the paper is sound. The authors add dynamic attention to better summarize the information on the visual feature map.
+ The presentation is mostly clear.
+ The experimental results support the claims.

**Weaknesses:**

Overall, the paper lacks significant enough contribution to vision and language community:
- The so called "interactive perception model" is actually a standard and common technique used everywhere. Early since attention was proposed, the visual information is dynamically summarized (attended). Some old works on VQA, such as NMN, have extensively used instruction-based condition to process visual information.
- The network/module themselves are also pretty common, and do not convey any significant "direction-shifting" messages.

The paper's writing also needs some improvements. For example, section 3.1 gives way too many engineering details (which should've been put into supplementary) and lacks top-down summarization for easier read.

**Questions:**

See above.

---

> ### Author Response · Authors · 2023-11-15
> **Respond to the comments of Reviewer c2o8**
>
> Thanks for your comments. We will respond to your comments below.
>
> **Q1**: The so called "interactive perception model" is actually a standard and common technique used everywhere.
>
> **A**: While the interactive perception is a concept that has been used in various research domains, our implementation within LMEye exhibits several distinctive and innovative aspects, particularly in its integration with Large Language Models (LLMs).
>
> **LLM-Centric Approach**: LMEye is uniquely engineered to function synergistically with LLMs. Here, the LLM serves as a central processor, orchestrating a dynamic and efficient interaction with the external visual environment. This is a significant deviation from previous models, which predominantly focused on static visual summarization.
>
> **Dynamic Interaction**: Unlike the methods you mentioned, such as NMN in VQA, LMEye empowers LLMs to actively initiate, gather, and process visual information in real-time, in response to ongoing human instructions. This approach aims to advance the frontiers of AI-Generated Content (AIGC) research by fostering interactions that are more aligned with human intentions and responsive to changes in the environment.
>
> **Performance and Efficiency**: Our comprehensive experimental evaluations demonstrate that LMEye not only significantly enhances zero-shot performance across various multimodal benchmarks but also accomplishes this with fewer parameters than other advanced multimodal large models. This indicates a substantial improvement in both practical effectiveness and efficiency, setting LMEye apart from existing methodologies.
>
> **Q2**: The network/module themselves are also pretty common, and do not convey any significant "direction-shifting" messages.
>
> **A**：
> Based on your comments, I would like to clarify and emphasize the innovative aspects of LMEye.
>
> **Clarification on Network/Module Commonality**: `It's essential to specify which aspect of the network or module is deemed 'common.', which makes us confused`. LMEye indeed builds upon foundamental large language and vision models. However, the uniqueness lies not in the individual components but in their novel integration and application.
>
> **Novel Bridging Approach**: The core contribution of LMEye is its innovative methodology to bridge a significant gap in the existing framework of Large Language Models (LLMs) and the visual environment. Unlike conventional methods that input visual information in a static manner and depend on a pre-tuned attention mechanism, LMEye introduces a dynamic and interactive approach RVII. This allows for the integration of visual information in real-time, responding adaptively to a range of human queries.
>
> **Dynamic vs. Static Processing**: This dynamic processing contrasts sharply with the common static input models. LMEye's approach is not just about focusing on key visual information through attention mechanisms but about continuously updating and integrating this information based on evolving human instructions.
>
> **Empirical Evidence of Superiority**: Our experimental results substantiate the effectiveness of LMEye. They demonstrate improved performance over existing methods, with less parameters.

---

### Official Review · Reviewer_E4sy · 2023-11-10

**Soundness:** 3 good
**Presentation:** 2 fair
**Contribution:** 3 good
**Rating:** 6
**Confidence:** 4

**Summary:**

This paper introduces LLMEye, a trainable module designed to enable dynamic interaction between LLMs and external vision information. LLMEye employs a two-stage approach for enhanced interaction. The first stage involves feature alignment training, which utilizes a Q-former from BLIP2 and linear projection layers to map image features into text space, capturing static visual information. In the second stage, LLMEye introduces a linear layer and a multi-layer transformer block to facilitate request-based visual information interaction. The paper also presents an evaluation of LLMEye on various multimodal benchmarks, including MME, SEED-Bench, and VQA tasks, alongside a custom evaluation set focusing on VQA tasks with long answers and detailed descriptions.

**Strengths:**

- The paper presents a variety of evaluation benchmarks and shows the potential of LLMEye
- Extending the capabilities of Multimodal-Models by training lightweight modules is an interesting direction also proposed by previous work (e.g MiniGPT4)

**Weaknesses:**

- Figure 1 could be improved a lot. First, I will suggest including and specifying each component of LMEye. Adding this will help the reader to connect the notation mentioned in Section 3.1 and the flow of the figure
- Ablations.
	- It would be beneficial to understand the contribution of the RVII module if authors include ablations with and without that module in MME and SEED-Bench benchmarks.
	- I also suggest specifying the Vision Model and Language Model with their corresponding parameters to have a better and fairer comparison while reading Tables 1 and 2.
	- Add ablations without RVII module in LLMEye (BLIP2) in Tables 3 & 4
	- Why not include the Qwen-VL model? I believe it was already published at the moment of the paper submission.
- Minor:
	- > ... which shows supervisor performances on various multimodal scenarios. (Page 2)

		Should it be "superior"?

**Questions:**

Please see above weaknesses

---

> ### Author Response · Authors · 2023-11-15
> **Respond to the valuable advice and comments from Reviewer E4sy**
>
> Thank you for your valuable suggestions and comments, which will undoubtedly enhance the quality of our manuscript. Due to page limitation, we will use two pages to respond to your comments.
>
> **Q1**: Suggestions for Improvement in Figure 1 and Writing.
>
> **A**: We appreciate your feedback on Figure 1 and acknowledge the need for its enhancement. To address this, we plan to revise Figure 1 to more clearly delineate and label each component of the LMEye system, ensuring that it accurately reflects the architecture and functionality discussed in the text. This revision will aim to enhance the figure’s clarity and its utility as a visual aid for readers. Additionally, we recognize the importance of clear and correct writing for effective communication of our research. To this end, we will conduct a thorough review and revision of the manuscript to correct grammatical errors, including the specific instance you mentioned on page 2.
>
> **Q2**: Ablations in MME and SEED-Bench benchmarks; Ablations without RVII module in LMEye (BLIP2) in Tables 3 & 4.
>
> **A**：In our prior paper, we highlighted the efficacy of the RVII module through a comparative analysis involving LMEye (BLIP), InstructBLIP, and BLIP, based on FlanT5-XL. InstructBLIP, by integrating multimodal instruction data to fine-tune the Q-former from BLIP, demonstrated superior performance across various tasks compared to BLIP itself. It's essential to note that the structural composition of InstructBLIP and BLIP remains unchanged. However, LMEye (BLIP) introduces a distinct component known as the Request-based dynamic visual information interactive module (RVII). While trained with instruction data akin to InstructBLIP, LMEye (BLIP) without RVII essentially aligns with the architecture of either BLIP or InstructBLIP. By juxtaposing the performances of these variations in Tables 1, 2, and 3, we can ascertain the specific impact of the RVII module.
>
> To address your constructive feedback, we conducted experiments by training LMEye (BLIP) without the RVII module using our code and instruction data. The subsequent experimental results, outlined in the following table, present  the significance and role of the RVII module in enhancing performance. All models are based on the same vision and language models (FlanT5-XL).
> | Models(FlanT5-XL) | MMBench.Test | SEEDBench | VCR  | VQA  | OK-VQA |
> |-------------------|--------------|-----------|------|------|--------|
> | BLIP-2            | -            | 49.74     | 57.3 | 59.95| 48.91  |
> | InstructBLIP      | 33.9         | 57.8      | 53.55| 62.56| 50.12  |
> | LMEye(ours) w/o RVII | 47.79   | 54.84     | 53.51| 56.72| 49.6   |
> | LMEye(ours)       | **62.6**         | **59.7**      | **57.5** | **63.2** | **54**     |
>
> **Q3**：Vision Model and Language Model with their corresponding parameters.
>
> **A**：Thank you for your valuable suggestion. We have collected relevant data as shown in the table below. We will also update specific parameter statistics in the next version.
> | Method           | Language model | Vision model        |
> |------------------|----------------|---------------------|
> | MiniGPT-4        | Vicuna 7B      | EVA-G 1.1B         |
> | MMGPT            | LLaMA 7B       | CLIP VIT-L/14 428M |
> | PandaGPT         | Vicuna 13B     | ImageBind ViT-H/14 1.2B |
> | VisualGLM        | ChatGLM 6B     | EVA-CLIP 1.1B      |
> | InstructBLIP     | Vicuna 7B      | EVA-G 1.1B         |
> | LLaVA            | LLaMA 7B       | CLIP VIT-L/14 428M |
> | LLaMA-Adapter-v2 | LLaMA 7B       | CLIP VIT-L/14 428M |
> | G2PT             | LLaMA 7B       | ViT-G 1.1B          |
> | mPLUG-Owl        | LLaMA 7B       | CLIP VIT-L/14 428M |
> | Ottar-I          | LLaMA 7B       | CLIP VIT-L/14 428M |
> | Shikra           | Vicuna 7B      | CLIP VIT-L/14 428M |
> | Qwen-VL           | Qwen 7B      | ViT-G/16 1.1B |
> | LMEye            | FlanT5-xl 3B   | ViT-G 1.1B     |

---

> > ### Author Response · Authors · 2023-11-15
> > **Continue to response.**
> >
> > **Q4**：Why not include the Qwen-VL model? I believe it was already published at the moment of the paper submission.
> >
> > **A**：Thank you for highlighting the significance of the Qwen-VL model. We understand its recent release and the advancements it brings, particularly through its multi-stage training methods. Initially, our study focused on established models available at the outset of our research. However, we acknowledge the importance of considering Qwen-VL within the context of our work.
> >
> > In response to your insightful suggestion, we have conducted a comparative performance evaluation in the MMBench and SEED-Bench benchmarks. Our findings reveal noteworthy performance differences: MMBench-Test: 61.8 (Qwen-VL) vs. 62.6; MMBench-Dev: 60.6 (Qwen-VL) vs. 65.5; SEEDBench-Img: 62.3 (Qwen-VL) vs. 59.7. Particularly in SEED-Img, Qwen-VL exhibits advantages over many existing large models, including our own. Moreover, a thorough examination of Qwen-VL's technical report has provided us with advanced insights and methodologies. We intend to integrate these into our variant model, LMEye (Qwen), leveraging Qwen-VL's architecture and training strategies. We believe this integration will significantly augment our research, enhancing the capabilities of our model and contributing substantially to its overall performance. In the revised version of our paper, we will include a detailed analysis and discussion of Qwen-VL's performance, showcasing the potential impact on the research landscape.

---

### Meta-Review · Area_Chair_i9wC · 2023-12-08

**Metareview:**

The paper presents LMEye, a model architecture that takes natural language queries and images to output language answers like VQA. There are two major steps in the process. In the first stage, LMEye encodes visual information that is then mapped to the token space of an LLM. The visual and language tokens are then sent to LLM 1 to produce visual requests, which along original visual encoding is sent to a "Request-based Visual Information Interaction module" to produce requested visual information. Finally the language query, the requested visual information and the original visual representation are processed by another LLM to produce the final answer. Overall, I see it's another attempt for multimodal VQA. Although the paper claims the contribution of dynamic interaction with visual information and human-like behaviors, the presented architecture doesn't show designs that adequately achieve these goals.

The reviewers generally like the direction of the work, e.g., "The motivation and implementation are clear and straightforward. The visual information should interact with language queries in multimodal LLMs." Reviewer DT9V. However, the reviewer opinions are quite split towards the novelty of the work. On the positive side, Reviewer RpWw thinks "The LMEye approach stands out as a monumental stride in the integration of visual perception with LLMs. The unique design of Request-based Visual Information Request (RVII), allowing LLMs to 'request' specific visual information, is both intuitive and groundbreaking, which reflects a deep understanding of how multimodal processing might emulate human brain-like cognition." On the opposite side, Reviewer c2o8 mentioned "The so called "interactive perception model" is actually a standard and common technique used everywhere. Early since attention was proposed, the visual information is dynamically summarized (attended)." While Reviewer E4sy and Reviewer DT9V are on the fence, they both pointed out the lack of in-depth ablation and analysis of the RVII module, which is at the core of the contribution. Reviewer E4sy further pointed out that "there is no essential difference between RVII and LoRA. LoRA is too insert some parameters inside the LLM , while RVII is more like an Adapter module outside LLM."

During the reviewer-AC discussion, Reviewer c2o8 further elaborated on their concerns:
> The idea of "LLM serving as central processor" is not really delivered. The authors claim dynamic but the model does not try more rounds of LLM thoughts generation. This should've been a basic ablation version of the model where multiple rounds of thoughts are generated and processed. Multi-round should be ablated and reported.

All in all, AC feels that the work has brought forward a few interesting and exciting concepts such as LLMs as the CPU. However, there lacks fundamental breakthroughs in the design of the architecture. In addition, the learning / testing setting doesn't really manifest the dynamic interactive aspect that the paper tries to achieve.

**Justification For Why Not Higher Score:**

The technical novelty of the work is limited and over-claimed.

**Justification For Why Not Lower Score:**

The overall direction of the work is exciting.

---

### Decision · Program_Chairs · 2024-01-16

Reject